# Acceptability testing of the Carers-ID intervention to support the mental health of family carers of people with profound and multiple intellectual disabilities

Mark Linden[1], Maria Truesdale[2], Rachel Aine Leonard[1]*, Michael Brown[1], Lynne Marsh[1], Stuart Todd[3], N. Hughes[4], Trisha Forbes[1]

1 School of Nursing and Midwifery, Queen's University Belfast, Belfast, Northern Ireland, 2 School of Health and Wellbeing, University of Glasgow, Glasgow, Scotland, 3 School of Care Sciences, University of South Wales, Caerleon, Wales, 4 Department of Sociological Studies, University of Sheffield, Sheffield, England

☯ These authors contributed equally to this work.

* rachel.leonard@qub.ac.uk

## Abstract

### Background

Providing care and support for a person with intellectual disabilities can be challenging and may negatively impact on family carers' health and wellbeing. A online support programme was co-designed with charitable organisations and family carers, to help meet the mental health and wellbeing needs of family carers.

### Objective

To test the acceptability of a newly developed online support programme for carers of people with profound and multiple intellectual disabilities.

### Methods

A sequential mixed-methods explanatory design was utilised. An adapted version of the Acceptability of Health Apps among Adolescents Scale was distributed to family carers across the United Kingdom and Ireland who had viewed the Carers-ID.com intervention. Participants were then invited to take part in an online interview. Qualitative and quantitative data were analysed separately and then brought together through the triangulation protocol.

### Results

Seventy family carers (47 female, 23 male) responded to the acceptability survey, with 10 (7 female, 3 male) taking part in interviews. Carers expressed high levels of programme acceptability (mean = 75.43 out of 88). Six themes were generated from interviews with family carers; i) time is precious, ii) the breadth and depth of module content, iii) it was somebody's experience; it was meaningful, iv) won't work for everyone, v) representation: people I could identify with, and vi) module specific suggestions for future changes. Based on our

**Data Availability Statement:** All anonymised data files are available from the UK data archive database (Doi: 10.5255/UKDA-SN-856210).

**Funding:** Funding was received from the UKRI ESRC REF: ES/W001829/1.The funders had no role in study design, data collection and analysis, decision to publish, or preparation of the manuscript.

**Competing interests:** The authors have declared that no competing interests exist.

**Abbreviations:** AHAAS, Acceptability of Health Apps among Adolescents Scale; ID, Intellectual disabilities; PC, Personal computer; PMID, Profound and multiple intellectual disabilities; UK, United Kingdom.

triangulation, four areas of convergence were identified: programme usability and ease, attitudes towards the programme, perceptions of effectiveness, and programme relatability.

## Conclusions

To be acceptable, online interventions for carers of people with intellectual disability need to be accessible, understandable and easy to use, as carers' free time can be limited. It would be important to investigate the effectiveness of online interventions for family carers, specifically considering which carers the intervention works for, and for whom it may not.

## Introduction

Research has consistently shown that caring for a person with intellectual disabilities is challenging [1]. Although rewarding [2], it can negatively impact on carers' health and wellbeing [3, 4]. The COVID-19 pandemic in 2019, and the public health preventative measures imposed to stop the spread of the virus (e.g. social distancing, travel restrictions, lockdowns, quarantine, and closure of public places), had a significant impact on the lives of families caring for a relative with intellectual disabilities [5, 6] and more so for those caring for a person with a profound and multiple intellectual disability (PMID) who tend to have greater support needs [7].

Whist there is not a universally agreed definition of PMID, individuals with PMID have both a profound intellectual disability and severe-to-profound motor disabilities [8, 9]. This is combined with limitations in adaptive behaviour [8] and communication [9]. In addition, they often have complex medical health needs and require support related to health conditions such as epilepsy, gastroesophageal reflux disease, respiratory infections and constipation [10, 11]. According to Schalock et al. [8], the support needs of persons with PMID can be considered as pervasive.

Much of the research evidence to date has focused on the effects of the pandemic on family carers of people with intellectual disabilities [12] with little attention focussing on the lived experiences of carers of people with PMID during the pandemic and their support needs. Research has shown that carers of people with intellectual disabilities experienced many challenges during the pandemic including the increased burden of the demands of caring due to the withdrawal of services and reduced support from family and friends [5, 6, 13]; increased mental health issues including anxiety, stress, loneliness, and depression [14, 15]; and social issues such as financial challenges and the loss of daily routines [16].

People with PMID mostly live at home with family members and rely heavily on their families for most of their care and support needs. Within the UK and Ireland, restrictions to contain the spread of COVID-19 have significantly impacted on these carers putting them at increased risk of stress and need for support given their reliance on professional and informal support [17, 18]. Whilst most of the research focuses on how the pandemic effected family carers of people with intellectual disabilities, limited research exists focusing on the effects and experiences of families with a family member with PMID and the longer term, post-COVID-19 outcomes for family carers. However, it is likely that they will require targeted support to assist with the short and longer term consequences of living through the pandemic. In 2022, Linden et al. [19] sought to capture the lived experiences of this population in a UK and Ireland wide study and to inform the development of an online support programme. They conducted thirty-two focus group interviews with 126 family carers and showed that the negative impacts of caregiving on carers mental health and wellbeing were exacerbated due to the

pandemic. Carers reported that an online support programme might prove useful in improving such impacts. Subsequently, online focus groups were held with key stakeholders including family carers of people with PMID and third sector and charitable organisations across the UK and Ireland, following the pandemic to co-design an online support programme, currently known as Carers-ID. This programme consists of fourteen modules covering topics including how to manage stress, reducing anxiety and low mood, promoting family relationships, accessing local support and future care planning. The aim of this study was to determine the acceptability of Carers-ID, among family carers of people with PMID. This study will focus on how carers engaged with and understood the content of the programme. It is also intended to inform further development of the programme and how this might be used to improve the health and wellbeing of carers of people with PMID. Given the dearth of research into online support interventions for carers of people with PMID, this study makes an important contribution to the existing knowledge base.

## Methods

### Design

This research utilised a sequential mixed-methods explanatory design to examine the acceptability of the newly developed programme with family carers. Our co-design team consisted of five voluntary sector organisations which represented the interests of family carers in each of the five countries of the UK and Ireland. Some were also self-advocates meaning that they were themselves carers. One further member of the co-design team was a family carer who did not also work for a voluntary sector organisation.

### Participants

Based on previous acceptability studies [20–24] we estimated we will need to recruit approximately 50 family carers. Carers (n = 70) who had undertaken the Carers-ID programme completed an acceptability questionnaire (See Table 1). The majority of participants were female (n = 47, 67%), with 23 male carers also taking part. Participants lived in England (n = 30, 43%), Scotland (n = 14, 20%), Wales (n = 13, 19%), Northern Ireland (n = 12, 17%), and the Republic of Ireland (n = 1, 1%). In relation to their caring role, participants were mostly mothers (56%, n = 39). Other roles included fathers (n = 17, 24%) and grandparent (n = 3, 4%). The mean age of participants was 40.79 years (SD = 12.77), with a minimum age of 24 years and a maximum of 65 years.

Carers (n = 10) agreed to take part in semi-structured, in-depth interviews to further explore acceptability and expand upon findings from the survey. Seven participants were female, nine were parents, with one sibling carer. Three carers were resident in Scotland, two were from Northern Ireland, two were from the Republic of Ireland, two were from England and one was from Wales (see Table 2).

### Measures

An existing questionnaire (Acceptability of Health Apps among Adolescents—AHAA Scale) intended to examine the acceptability of health Apps [25] was adapted to focus on an online programme. Some wording of items were changed to reflect "online programmes" as opposed to the original "app", the overall structure of the original scale was not changed. For example, items which originally stated 'I like this App' or 'This App is confusing' or 'Eating healthy is unimportant to me' were altered to read 'I like this programme', 'This programme is confusing', 'Disability is unimportant to me'. The original and adapted questionnaires comprised 22

**Table 1. Demographics of carer responses to the survey.**

| N = 70 | | |
|---|---|---|
| Sex | Female | 47 (67%) |
| | Male | 23 (33%) |
| Carer | Father | 17 (24%) |
| | Mother | 39 (56%) |
| | Grandparent | 3 (4%) |
| | Other carer | 11 (16%) |
| Country | England | 30 (43%) |
| | Scotland | 14 (20%) |
| | Wales | 13 (19%) |
| | Northern Ireland | 12 (17%) |
| | Ireland | 1 (1%) |
| Family size | 2 | 5 (7%) |
| | 3 | 23 (33%) |
| | 4 | 35 (50%) |
| | 5 | 4 (6%) |
| | 6 | 1 (1%) |
| | Missing data | 2 (3%) |
| Age (years) | Mean (SD) | 40.79 (12.77) |
| | Minimum—Maximum | 24–65 |
| | Missing data | 9 |

questions consisting of 6 subscales. These comprised Affective Attitude, Burden, Ethicality, Intervention Coherence, Perceived Effectiveness and Self-Efficacy. Participants responded on a four-point Likert scale ranging from 1 strongly disagree to 4 strongly agree. Negatively

**Table 2. Demographics of carers who participated in interviews.**

| N = 10 | | |
|---|---|---|
| Sex | Female | 7 (70%) |
| | Male | 3 (30%) |
| Carer | Parent | 9 (90%) |
| | Sibling | 1 (10%) |
| Country | England | 2 (20%) |
| | Scotland | 3 (30%) |
| | Wales | 1 (10%) |
| | Northern Ireland | 2 (20%) |
| | Ireland | 2 (20%) |
| Family size | 2 | 1 (10%) |
| | 3 | 6 (60%) |
| | 4 | 2 (20%) |
| | 5 | 1 (10%) |
| Age (years) | 25–34 | 1 |
| | 35–44 | 3 |
| | 45–54 | 3 |
| | 55–64 | 3 |
| | Minimum—Maximum | 35–64 |
| | Missing data | 0 |

**Table 3. Interview guide for acceptability testing.**

| |
|---|
| 1. Having used the programme, what did you think of it? |
| *Potential prompts*: *Can you tell me more about that*? *Can you give me an example of that*? |
| 2. Can you tell me about any parts of the programme that you thought worked well? |
| *Potential prompts*: *Can you tell me more about that*? *Can you give me an example of that*? |
| 3. Can you tell me about any parts of the programme that you thought really didn't work? |
| *Potential prompts*: *Can you tell me more about that*? *Can you give me an example of that*? |
| 4. Is there anything about the programme that you would like to change? |
| *Potential prompts*: *Can you tell me more about that*? *Can you give me an example of that*? |
| 5. Is this a programme that you would use? Can you explain your answer? |
| 6. Is there anything that we have not talked about today that you would like to talk about before we finish the interview? |

worded items on the scale were reverse scored. A total score was calculated by adding the scores on each subscale. Participants could score a minimum of 22 or a maximum of 88 with higher scores indicating a greater degree of acceptability. The validity of the AHAA scale was established through confirmatory factor analysis and was also assessed for internal consistency ($\alpha$ = 0.91) [25]. Internal consistency of the adapted version of the questionnaire used in the present study was determined by Cronbach's alpha as 0.93.

Questions for the semi-structured interviews sought to explore participants' experiences of using the programme in addition to their thoughts on its acceptability. Six questions, with accompanying prompts, were developed with the research and co-design teams. The interview schedule can be found in Table 3.

## Procedures

Prior to commencing data collection ethical approval was sought and granted from a University ethics review board (Ref: MHLS 21_38) at the lead author's institution. Informed written consent was gathered from participants prior to completing the study. Recruitment was facilitated through our five voluntary sector partner organisations. Potential participants who had previously taken part in focus group interviews about the impact of the COVID-19 pandemic on carers [19] were invited to view the programme and complete the acceptability questionnaire. Participants were emailed a letter of invitation to inform them of the research followed one week later by an information sheet and a link to the questionnaire. Carers who decided to take part were required to read the information sheet and provide their written consent to take part. MS Forms was used to collect data on participant demographics (age, sex, family size etc.) and acceptability. Recruitment began on in October 2022 and end in January 2023.

Those who were invited to complete the questionnaire were asked to indicate whether they would be willing to take part in an online interview (using the Zoom platform) to explore their views in greater depth. Two members of the research team conducted the interviews which were audio recorded and transcribed verbatim for analysis. Interviews lasted from between 16 to 63 minutes. Participants who completed the survey received a £10 voucher following completion. Those participants who also completed interviews, they received a further £20 in thanks.

## Data analysis

Means and standard deviations, and medians and interquartile ranges were calculated for each subscale and total score on the AHAA, determined by adding all subscale scores. Participants

total scores could range between a minimum of 22 or a maximum of 88, with higher scores indicating a greater degree of acceptability. Independent t-tests were conducted to compare the mean difference between sex and carer role on acceptability, with one-way analysis of variance (ANOVA) employed to compare the mean difference between carer family size, and country of residence, on programme acceptability. Effect sizes were estimated using Cohen's d for t-test and partial eta squared ($\eta^2$) for ANOVAs to convey the difference in magnitude between groups. Levels of significance (small, medium and large) were determined through guidance by Cohen [26]. Post-hoc tests using the Tukey correction were then used to identify the difference between variables with more than two levels (e.g. country and carer family size). All analyses were carried out in SPSS Statistics for Windows [27].

Qualitative data from online interviews were audio recorded, transcribed verbatim and analysed by inductive thematic analysis [28]. The six steps set out by Braun and Clarke [28] were followed; familiarisation with the data; generating initial codes; searching for themes; reviewing themes; defining and naming themes; and producing the report. An inductive approach refers to the development and modification of codes throughout the coding process. Analysis of qualitative data was conducted by three members of the research team. Authors, ML, RL, and TF, coded transcripts and analysed all codes to ensure they were meaningful and coherent and reflected content of the transcripts [29]. The research team met regularly to ensure rigour throughout the analysis process and to further develop and refine themes [29].

Qualitative and quantitative data were synthesised according to the triangulation protocol developed by Farmer et al., [30] to provide a deeper understanding of acceptability than that provided by a single approach. The protocol allows for the synthesis of data based on where it may agree, partially agree, complement, where there may be dissonance or silence [30, 31]. Full agreement occurs when both sets of data align on elements of comparison [30]; complementary synthesis occurs were data supports a perspective; dissonance refers to where data sources conflict and silence occurs where a concept is present in one dataset but not another [31]. A mapping matrix was used to provide a visual representation of convergence of the triangulated data. Qualitative themes and quantitative descriptive data were mapped onto the matrix, along with exemplar quotes to illustrate the meaning behind qualitative themes. The final part of the mapping matrix included the analytical integration which provided an overview of where the data agreed, complemented, diverged or had mixed findings (had both divergent and agreement).

## Results

Seventy carers completed the acceptability questionnaire (See Table 1). The majority of participants were female (67%, n = 47), parent carers (80%, n = 56), and came from England (43%, n = 30). The mean age of participants was 40.79 years (SD = 12.77), with a minimum age of 24 years and maximum of 65 years.

Participants had an overall mean score of 75.43 (SD = 9.97) on the AHAAS, ranging from 55 to 88 (See Table 4). Total mean scores on each subscale were also calculated for Affective Attitude (13.91, SD = 2.32), Burden (14.04, SD = 2.18), Ethicality (14.46, SD = 2.26), Intervention Coherence (10.44, SD = 1.82), Perceived Effectiveness (13.06, SD = 2.39), and Self-Efficacy (9.51, SD = 2.05).

Independent t-tests were conducted to compare mean differences between sex (male vs female) and carer role (mother vs father) on programme acceptability, as determined by total score on the AHAA (see S1 Table for descriptive data for each factor). Grandparents were omitted from this analysis due to lack of data (n = 3). There was no statistically significant difference in the level of programme acceptability between males and females (t(68) = -1.678,

**Table 4. Mean (SD), lowest and highest scores on six subscales and total score of AHAAS.**

|  | Potential Min—Max scores | Mean (SD) | Median (IQR) | Participant Min—Max scores |
|---|---|---|---|---|
| Affective Attitude | 4–16 | 13.91 (2.32) | 15 (12–16) | 8–16 |
| Burden | 4–16 | 14.04 (2.18) | 15 (12.75–16) | 7–16 |
| Ethicality | 4–16 | 14.46 (2.26) | 16 (13–16) | 8–16 |
| Intervention Coherence | 3–12 | 10.44 (1.82) | 11 (9–12) | 3–12 |
| Perceived Effectiveness | 4–16 | 13.06 (2.39) | 12.50 (12–16) | 8–16 |
| Self-Efficacy | 3–12 | 9.51 (2.05) | 9 (8–12) | 5–12 |
| Total score | 22–88 | 75.43 (9.97) | 75.50 (67–85) | 55–88 |

p = 0.10), or between mothers and fathers (t(54) = -1.743, p = 0.091). The effect size was small for both analyses, with a Cohen's d of 0.42 and 0.49 respectively.

One-way ANOVAs were conducted to compare the effects of country of residence and family size respectively, on programme acceptability (see S1 Table for descriptive data for each factor). One-way ANOVAs showed no statistically significant difference between country of origin (F(4,65) = 0.290, p = .883) and family size (F(4,63) = 0.331, p = .856) regarding programme acceptability. The effect size, eta squared ($\eta^2$), was 0.01 and 0.02 respectively, indicating a small effect.

## Qualitative results

Overall, ten carers completed interviews regarding programme acceptability. Seven were female and three were male, with nine being parent carers and one a sibling carer. Three carers were from Scotland, two came from Northern Ireland, two from the Republic of Ireland, two from England and one from Wales (see Table 2). Interview duration ranged from 16 to 63 minutes.

Six themes were generated from interviews with family carers (see Table 5). These included: i) "time is precious", ii) "the breadth and depth of module content", iii) "it was somebody's experience; it was meaningful", iv) "won't work for everyone", v) "representation: people I could identify with", and vi) "module specific suggestions for future changes". Each will be discussed in turn.

**i) Time is precious.** Participants highly valued the programme's ease of use and flexibility. This was important to participants as their time was their most precious resource. This theme is discussed below.

Participants enjoyed being able to "*jump in and out*" of the programme, which offered some flexibility of use. Further adding to the ease of use, was the format of the different

**Table 5. Themes and sub-themes from acceptability interviews.**

| Theme | Sub-themes |
|---|---|
| i. Time is precious |  |
| ii. The breadth and depth of module content |  |
| iii. It was somebody's experience; it was meaningful |  |
| iv. Won't work for everyone |  |
| v. Representation: people I could identify with |  |
| vi. Module specific suggestions for future changes | *1. Peer mentoring*<br>*2. Enhancing resilience*<br>*3. Future care planning*<br>*4. Language*<br>*5. Sharing the resource* |

modules and sections, which participants felt was "*easy to consume*" (Male, aged 45–54, Wales) and allowed participants to access the specific information they wanted quickly; "*I really liked it. I liked the way it was set out. The different sections in it were very clear. And you could go to whatever was relevant to you at that particular time*" (Female Carer, Aged 55–64, Northern Ireland). In addition, participants noted that the programme was easy to navigate for those who were not technical and worked well on both computer and phone formats. For carers this was important as many accessed support through their phone: "*It worked quite well with a phone, because I was worried. . . because I mostly access things like this on my phone. I don't have time to put on a PC or a laptop*" (Female Carer, Aged 35–44, England).

Furthermore, participants appreciated the way in which information was presented in the programme. Specifically noting the layout of modules: video, definition, further links and information boxes: "*And I think the way that it's divided into chunks, I think it's really easy to comprehend and understand. And then it's easy to delve deeper into the subjects afterwards*" (Female Carer, Aged 35–44, England).

**ii) The breadth and depth of module content.**   There were mixed findings in relation to the overall content of the Carers-ID programme, with some participants reporting a good breadth and depth of topics, while others felt information was too generic given the complexities of their caring responsibilities. All participants valued the video content, describing this as heartfelt and meaningful.

Some participants felt that the module content was comprehensive and covered a wide range of subjects, which provided useful information: "*to me it was very well thought out. As I say, the topics covered, I think. . . the siblings who are carers will relate to some people. So they tried to cover all angles. And I thought they offered. . . it was a wide range of advice that was there, that people could go on and get*" (Female Carer, Aged 55–64, Northern Ireland).

For others, the programme lacked detail in places, and was too generic: "*sometimes I felt connected, like the author was speaking directly to me. And then it switches to be generic, like it's talking in a wider, generic sense, but talking to people who don't know what being a carer is*" (Female Carer, Aged 35–44, Scotland). In addition, some participants felt that the information provided through the programme was already readily available online: "*the website as it stands . . . I can click into Google and find a lot of the stuff that is there. I have been down those roads. I have looked up this stuff*" (Male Carer, Aged 55–64, Northern Ireland).

Within the content of the Carers-ID programme, there are tips and advice throughout the 14 modules. Participants had mixed views on the advice offered. Some participants valued the tips and advice, stating that carers often struggled to ask for help: "*practical tips are very much helpful, because some people don't want to ask others . . . they don't want to share their feelings, thoughts, to others. So yeah, tips, some practical tips are also helpful*" (Female Carer, Aged 35–44, England). Participants also acknowledged that even if tips and advice were not needed at that particular time, the ease of navigation of the programme allowed carers to access advice when needed: "*there were hints and tips about how to deal with that particular situation, that you may or may not find helpful, but they were good to be there. And even if they didn't appear helpful at that time, the next time you go onto it, it possibly could. I really liked it*" (Female Carer, Aged 55–64, Northern Ireland).

Other participants felt that the programme had too much signposting to external charities or websites, and not enough specific tips and advice: "*otherwise I'm just going to read through it and just go, this hasn't actually helped me, and it's just another charity or whatever, that's signposting me round in circles and not actually doing anything to support me*" (Female Carer, Aged 35–44, Scotland). In addition, some participants wanted more specific advice that was relevant to their situation. For example, some participants suggested having advice on filling in benefit forms, advice on day centres, and practical tips to reduce low mood: "*some of them really make*

*me want to go, oh*! *Like I want to reduce my mood. How would I do that*? *That's why I would click on the link. I'll go, right, how can I make my mood better*?*"* (Female Carer, aged 35–44, Scotland). Furthermore, participants felt that the advice was good, however was not always feasible for all carers in their individual circumstances. For example, as one participant described their issues with the advice provided within the sleep module: *"it's all well and good telling me how to get better sleep*, *but I can't get better sleep because I have to have a light sleep so that I can react if [name] has a seizure. There may be people that these modules would help. But not me*" (Male, Aged 55–64, Northern Ireland).

**iii) It was somebody's experience; it was meaningful.** The Carers-ID programme provided video content within each module. The videos depicted carers recounting their experiences of each of the 14 topics. For example, the peer support module contained a video of carers speaking about their experience of support and the importance of peer support for carers. All participants valued the videos within the programme. Participants described the videos as heartfelt and meaningful and reported learning from the videos and other carers: "*but the other videos*, *even though they were sad because it was somebody's experience, it was meaningful*" (Female Carer, Aged 55–64, Scotland).

Participants also liked that the videos related well to the modules, for example, where a video clip was talking about stress, below that video was a definition of stress and some resources: "*what I liked was the video clip with the definition. And when I looked down, that definition of stress*, *that was me. Those physical consequences and symptoms of stress. And then I looked beneath and I saw links to places that could help. So, I thought that is a nice, compact package*" (Male Carer, aged 45–54, Wales).

Some participants suggested that videos could be enhanced by incorporating professional input, particularly for modules such as managing stress, sleep and nutrition.

**iv) Won't work for everyone.** Participants who had been caring for many years had concerns that the programme may be better suited to those new to their caring role. Participants with a wealth of caring experience expressed views that the programme may not offer any new information for them: "*they are not telling me anything I don't know. Been there*, *done that*, *got the T-shirt*" (Female Carer, Aged 55–64, Scotland). Some participants reported already knowing a lot of the information and resources within the programme: "*For me, I already knew*, *because my son is diagnosed with autism for five years. So, actually I search a lot, I got to know a lot from other parents or other people. So*, *some of the services and support I already know about*" (Female Carer, Aged 35–44, England). For these participants, they struggled to see how the programme would meet their particular needs: "*but I am genuinely not sure how it would address any of the particular needs that I have as a carer, and any of the struggles that I have as a carer. It led me to agencies*, *organisations*, *NGOs that I have had experience of before, and found often to be a dead end*" (Male, Aged 55–64, Northern Ireland).

However, participants acknowledged that the programme would undoubtedly be helpful for carers who are at the beginning of their caring journey or new to their caring role. Participants stated that having this programme at the beginning of their caring journey would have been comforting and would provide support through the information and advice offered: "*Just somebody starting on their journey I think would find it comforting, and that's probably as much as we can hope for. Just to give them a bit of strength*" (Female Carer, Aged 55–64, Scotland). In addition, participants stated that the programme would support carers in those first few years of their caring role, at a time when carers often feel lost, confused and hopeless: "*But I think what you have is really, really good for people that are in the first three years*, *that are lost. That are maybe in that initial diagnosis stage, that are maybe suffering from depression, which we all got into. You know, the shock of it and all that kind of thing*" (Female Carer, Aged 45–54, Republic of Ireland).

**V) Representation: People I could identify with.** Participants spoke about the representation of different carers within the programme. The majority of participants appreciated the diversity and representation of carers from different regions and the representations of male carers. Participants felt that the research team had listened to carers in the development of the programme and translated those findings into something meaningful: "*You guys have done a wonderful job trying to get information from all of us that did the focus group and tried to make something meaningful out of it*" (Female Carer, Aged 35–44, England). Participants described the value of hearing male voices within the programme as they were often underrepresented: "*and it was nice that you did focus on male. . . fathers and brothers and whoever, rather than. . . Because whenever I go to these things, it's always women*" (Female Carer, Aged 35–44, England). Furthermore, participants appreciated having carers from different regions of the UK represented within the programme: "*I was also pleased to hear regional voices. Accents that I could recognise as being my own. People I could identify with*" (Male carer, Aged 55–64, Northern Ireland). However, one participant felt there was a lack of representation and diversity in the videos, and a lack of recognition of carers in a variety of caring roles, specifically young carers, carers from ethnic minorities and carers with disabilities.

**vi) Module-specific suggestions for future changes.** Some participants had specific feedback about individual modules and future changes. These comments related to modules on enhancing resilience, peer mentoring, future care planning, language within the programme, and sharing the resource. Each are presented below.

*Peer mentoring.* There was consensus among the participants that the peer mentoring module would be helpful for all carers. Peer support and social connection was acknowledged by the participants as fundamental for carers. Participants described taking part in the programme with the aim of developing social connections: *"we talk about things like social isolation and loneliness and stuff like that. So maybe some carers as well are coming on, looking for that connectivity element too"* (Female Carer, Aged 35–44, Scotland). Participants valued the peer mentoring module, described the benefits of learning from other carers, hearing the varied experiences of other carers, and connecting with others. For the male participants, the peer support element of the programme held additional value, given the challenges of male support: "*and as I say, in that male sphere, it's very hard to be like that with other men. Just to have a person that I could go to from time to time and give off stink a bit, who I know understands some of this stuff without me having to explain*" (Male Carer, Aged 55–64, Northern Ireland).

Participants also recognised the potential challenges for the mentor, discussing the emotional burden of mentoring, access to appropriate support and training for the mentoring and the recognition that the mentor is there for support, not to provide solutions: "*I like the idea of having someone to talk to, someone to turn to when everything collapses. In the full understanding that that person will not solve anything*" (Male Carer, Aged 55–64, Northern Ireland).

*Enhancing resilience.* Some participants reported being confused about the term "resilience", whilst others felt it was an important concept, but difficult to express without using jargon. Others felt that the word resilient should not be used in the context of caring. Participants also noted that the term resilience is often overused and misunderstood, which creates challenges in talking about this concept. Some participants were not sure what resilience was and felt it needed to be better defined within the module. Another participant, acknowledged that resilience was an important concept, however, they felt it was hard to convey the meaning without using technical language: "*and I think the trouble is that resilience is such an important idea and word, that I don't know quite how you convey that, and reduce the use of jargon*" (Male Carer, aged 45–54, Wales).

*Future care planning.* The future care planning module evoked many emotions among participants.

Some participants wanted more specific information and advice, for example on developing a will or trust fund, how to make a formal complaint, or other legal advice regarding future care planning. One participant suggested input from legal professionals within this module: "*so some sort of concrete legal advice on the simple questions. Or even people that can be recommended down the line*" (Female Carer, Aged 45–54, Republic of Ireland). Without some of this more specific information, some participants felt that the module did not meet their needs: "but that future care planning module doesn't answer any of the stuff that I need answered" (Male Carer, Aged 55–64, Northern Ireland). Some participants expressed their difficulty in completing the future care planning module. For carers, thinking about the future had a significant emotional impact, particularly decisions around future care for their families when they were no longer able. Carers stated: "*a few of the modules hit hard, especially the future planning one. That is a difficult conversation. You must address it. And I admit I was crying. . . I think it's quite emotional for quite a few people reading it*" (Female Carer, Aged 35–44, England).

*Language*. Some participants had concerns about the language used in the programme. Some did not like the term "module", describing that it felt too academic or portrayed a sense of learning as opposed to support. Participants suggested using the title of each module, instead of the term "module". In addition, some participants felt that modules had different writing styles throughout, at times personal and at time academic. Some participants described being able to tell that modules were not written by carers: "*and I think that there were parts that felt almost a bit muddled. Like you could tell that it was written. . . that there was carer contribution, but you could tell that it definitely wasn't written by carers*" (Female Carer, Aged 35–44, Scotland).

*Sharing the resource*. Participants spoke about the importance of sharing this resource with a diverse audience, such as schools, health care professionals, other carers, and the public. Participants wanted to share the programme with a wide range of people with the aim of helping understand the caring role and sharing knowledge. As this participant described the benefits of sharing the programme with other parents and carers and charity groups: "*but I think parents at my school, my local parent carers group, they would really appreciate it. I think it's something that even signposting different charities or different organisations to*" (Female Carer, Aged 35–44, England).

## Triangulation of qualitative and quantitative results

The qualitative and quantitative data on its own provides part of the story of programme acceptability. Together, they contribute to a higher level of analysis and a broader understanding of the research question (Farmer et al., 2006). Based on our triangulation analysis of qualitative and quantitative data, four areas of convergence were identified, these included, programme usability and ease, attitudes towards the programme, perceptions of effectiveness, and programme relatability. Each of these areas of convergence are discussed in turn (see Table 6).

## Programme usability and ease

Participants easily understood the programme and how it worked (mean = 10.44), felt confident in their ability to engage (mean = 9.51) and perceived low burden of programme use (mean = 14.04). Interview findings confirmed these and further added to the interpretation. For example, participants expressed the value of the Carers-ID programme in being easy to use, and understandable because of the limited time they have to engage with it.

**Attitudes towards the programme.** The survey showed that participants held a positive affective attitude towards the programme (mean = 13.91). However, interview findings pointed to a combination of both positive and negative attitudes, offering a more nuanced

**Table 6. Triangulation of qualitative and quantitative results.**

| Qualitative—themes | Quantitative -descriptive data of subscales | Exemplar quote | Analytical integration |
|---|---|---|---|
| *"Time is precious"* | Intervention coherence. Potential min and max score (3–12) Median score 11 | "*And I think the way that it's divided into chunks, I think it's really easy to comprehend and understand. And then it's easy to delve deeper into the subjects afterwards*" (Female Carer, Aged 35–44, England). | Convergence<br>• Survey data indicated that participants had a high understanding of the programme and how it worked.<br>• Survey data suggested that participants felt confident that they could participate in the programme.<br>• Interview data converged with survey data, with participants suggesting the programme was easy to comprehend and use.<br>• From the survey, participants' scores indicated low perceived burden from the programme.<br>• Interview data also suggested that participants appreciated the ease of use of the programme given their most precious asset was time. Participants suggested some improvements to the programme to protect carers' time. |
| | Burden. Potential min and max score (4–16) Median score 15 | "*I suppose maybe something you might think of is, could you have maybe more summary, and then the long version? Some of us are very. . . we like reading. And maybe there's an option there to have a little bit more information in some of the areas*" (Female Carer, Aged 45–54, Republic of Ireland) | |
| | Self-efficacy. Potential min and max score (3–12) Median score 9 | "*I really liked it. I liked the way it was set out. The different sections in it were very clear. And you could go to whatever was relevant to you at that particular time*" (Female Carer, Aged 55–64, Northern Ireland) | |
| *"The breadth and depth of module content"* | Attitude Potential min and max score (4–16) Median score 15 | "*the website as it stands . . . I can click into Google and find a lot of the stuff that is there. I have been down those roads. I have looked up this stuff*" (Male Carer, Aged 55–64, Northern Ireland). | Mixed<br>• Participants in the survey had high scores in relation to affective attitude about the programme.<br>• Interview findings were mixed with some participants reporting good breadth and depth of topics, and meaningful experiences of other carers. Others felt information was too generic. |
| *"It was somebody's experience; it was meaningful"* | | "*but the other videos, even though they were sad because it was somebody's experience, it was meaningful*" (Female Carer, Aged 55–64, Scotland). | |
| *"Module specific suggestions for future changes"* | | "*You guys have done a wonderful job trying to get information from all of us that did the focus group and tried to make something meaningful out of it*" (Female Carer, Aged 35–44, England). | |
| *"Won't work for everyone"* | Effectiveness. Potential min and max score (4–16) Median score 12.50 | "*But I think what you have is really, really good for people that are in the first three years, that are lost. That are maybe in that initial diagnosis stage, that are maybe suffering from depression, which we all got into. You know, the shock of it and all that kind of thing*" (Female Carer, Aged 45–54, Republic of Ireland). | Complement<br>• Survey data suggested that participants believed that the programme was likely to achieve its purpose.<br>• Interview data further added to this by highlighting that participants felt the programme would be most effective for those at the start of their caring journey. |
| *"Representation: people I could identify with"* | Ethicality. Potential min and max score (4–16) Median score 16 | "*and it was nice that you did focus on male. . . fathers and brothers and whoever, rather than. . . Because whenever I go to these things, it's always women*" (Female Carer, Aged 35–44, England). | Complement<br>• Survey data indicated high perceived programme ethicality.<br>• Through the interviews, participants felt that the programme represented both female and male carers. |

interpretation of carers' attitudes. For example, carers expressed positive attitudes towards the videos of other carers sharing their experiences within the programme. However, some felt information provided in the programme was too generic, with advice and guidance not being practical for all carers given the complexities of their caring role. Taken together, this indicates that for some, the programme has meaning and offers beneficial advice, while other carers may want specific advice and information that is not currently offered.

**Perceptions of effectiveness.** The survey results suggested that participants believed that the programme was likely to achieve its purpose and be effective for family carers (mean score = 13.06). Comparing these findings with the interview analysis broadened this perspective. For example, while carers felt that the programme would be beneficial for carers, they believed that it would be particularly effective for those at the beginning of their caring journey, given this is often a time of isolation, confusion and information seeking.

**Programme relatability.** Quantitative findings on programme ethicality complemented the qualitative interview findings regarding the representation of carers within the

programme. Quantitative results indicated high perceived programme ethicality (mean = 14.46). Carers appreciated the diversity and representation of those from different regions, with different perspectives and experiences and the specific representations of male carers.

## Discussion

This study examined the acceptability of a newly developed programme for family carers of people with PMID, using a sequential mixed-methods explanatory design. Quantitative results suggested a high degree of programme acceptability (mean score = 75.43), with no differences evident on variables such as sex, country of origin, family size, and carer role. Qualitative findings added further understanding, suggesting that in order to be acceptable, online interventions for carers of people with intellectual disabilities need to be accessible, understandable, and easy to use, as carers' time is immensely valuable. Taken together, our triangulated results give a deeper understanding of programme acceptability. We found that acceptability was facilitated by the online intervention being relevant and meaningful to carers, whilst not being burdensome to complete. Findings also indicated that the intervention may not work for everyone and may be more valuable for those carers at the early stages of their caring journey.

Family carers found the content of the Carers-ID programme relevant and relatable. A study by Contreras et al [32], which explored the acceptability of an online intervention for carers of people with dementia, found that intervention acceptability was facilitated by content which was relatable to the carer's own needs and experiences. One element that participants found particularly relatable were the videos of other carers sharing their experiences. In addition, the videos and content reflected a diverse demographic of carers including underrepresented groups such as male carers. Barriers such as attitudes towards services, lack of awareness, language and cultural appropriateness can prohibit carers from underrepresented groups from engaging in some interventions [33]. Therefore, to encourage engagement, interventions should be reflective of the diverse demographics of family carers, particularly underrepresented groups such as male carers, ethnic minorities, siblings, use accessible language and take account of cultural variation.

Our study found that the Carers-ID online programme was perceived as easy to use (specifically the format and self-paced nature) and was not considered burdensome, indicating its acceptability among carers. The COVID-19 pandemic added to the burden of demands placed on family carers [5, 6, 13]. To be acceptable, online interventions for family carers cannot further add to this burden. Previous research among carers of people with long-term illnesses, similarly found that family carers appreciated the flexibility and self-paced nature of online interventions [34] given the busy nature of their lives. In addition, carers can struggle to access interventions due to time, family pressures, cost, and availability of services [35]. The online nature and ease of use offer an opportunity to overcome such barriers so that carers can engage and benefit from online interventions.

While our findings indicate good acceptability, participants felt that the programme may be more valuable to some carers than others. Given the vast range of experiences and unique situations of carers of people with PMID, an online intervention may not be able to provide all information for all circumstances. While many participants found the content of the intervention comprehensive, some carers wanted more specific information for their circumstances. In addition, findings indicated that the programme may be more beneficial for carers at the start of their caring journey. Previous research among carers of people with dementia in the UK also found that more experienced carers did not find the online programme as valuable as those in the early stages of their caring journey [36]. Therefore, there is a need for further

investigation of the relationship between programme effectiveness and carer experience. In addition, given that this current study has established acceptability, future research should now examine the feasibility and effectiveness of the Carer-ID intervention.

## Policy implications

Family carers often discuss the lack of support they receive from services and often experience this as a combative process. The fight for services may be dependent on location, funding availability, perseverance and knowing how to 'work the system'. More experienced family carers possess a wealth of knowledge which service providers should utilise to improve delivery of services in a manner which better meets their needs.

The nature of caregiving means that carers are resilient and dedicated. However, even the most resilient individual can experience low mood and may need emotional support from time to time. Voluntary sector organisations recognise the importance of connecting people for the purposes of providing support and new learning. The Carers-ID programme also uses peer support which carers greatly appreciated. This is a relatively low-cost intervention which service providers could easily implement to improve the well-being of carers.

Delivery of services via online platforms may offer an accessible, low-cost approach, which could better address the needs of carers. While the initial creation and refinement of an online service would require time and resources this could be used over many years and include large numbers of individuals. Of course, not all carers may have access to technology, or a reliable internet service, however, as technology improves, costs are reduced and uptake of online services will become more viable.

## Limitations

Our study included family carers from across the UK and Ireland and included a broad age range (24–65 years). However, our recruitment was less than anticipated in Ireland. As such, our findings are more representative of family carers in the UK. In the quantitative phases the small sample size limits the generalizability of the findings. However, through our recruitment efforts we made every attempt to make sure the sample was diverse and reflective of the carers population as possible. A third of participants in our survey were male who are usually under-represented in research on family carers. It is important to hear the voices of both female and male carers as they each perform caregiving duties within their families and face their own mental health and well-being needs. Our recruitment through five voluntary sector organisations may have resulted in the inclusion of more experienced and knowledgeable carers. The average age of our sample was almost 41 years indicating that these individuals had been providing care, and likely seeking information and services, for several years. This may explain why some carers felt the programme would be better suited to younger carers. small sample size for the quantitative work We triangulated our quantitative and qualitative findings using a well-established protocol which allowed for greater depth of understanding on acceptability of the Carers-ID programme. Conducting assessment using either of these approaches in isolation would have omitted important information on acceptability.

## Conclusions

The Carers-ID programme is acceptable to family carers of people with PMID. This study demonstrates how co-design can be used to create interventions or programmes with carers of people with intellectual disabilities (ID) which better address their needs. However, the programme may prove most useful for carers who are at the earlier stages of their caring role. As such, Carers-ID should be recommended by specialist ID teams, and other health and social

care professionals, to carers of younger children who are seeking information about ID. Provision of ongoing support for carers is needed if they are to continue to lead happy and healthy lives whilst providing crucial care to their families. Support could be delivered through healthcare providers, voluntary sector organisations or through peer support groups as in Carers-ID. Whatever the mechanism, it is important that carers feel that they are not alone and can turn to others at times of need.

## Supporting information

**S1 Table. Mean (standard deviation) and median (interquartile range) for sex, carer, carer family size, and country of residence, on programme acceptability total score.**
(DOCX)

## Acknowledgments

The authors thank all the family carers who took part in this study.

## Author Contributions

**Conceptualization:** Mark Linden.

**Data curation:** Rachel Aine Leonard.

**Formal analysis:** Rachel Aine Leonard.

**Funding acquisition:** Mark Linden.

**Investigation:** Rachel Aine Leonard.

**Methodology:** Rachel Aine Leonard.

**Project administration:** Rachel Aine Leonard.

**Supervision:** Mark Linden.

**Visualization:** Rachel Aine Leonard.

**Writing – original draft:** Mark Linden, Maria Truesdale, Rachel Aine Leonard.

**Writing – review & editing:** Mark Linden, Maria Truesdale, Rachel Aine Leonard, Michael Brown, Lynne Marsh, Stuart Todd, N. Hughes, Trisha Forbes.

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
