## [Decision Letter · Decision Letter 0]

9 Aug 2024

PONE-D-24-12377Acceptability testing of the Carers-ID intervention to support the mental health of family carers of people with profound and multiple intellectual disabilities.PLOS ONE

Dear Dr. Leonard,

Thank you for submitting your manuscript to PLOS ONE. After careful consideration, we feel that it has merit but does not fully meet PLOS ONE’s publication criteria as it currently stands. Therefore, we invite you to submit a revised version of the manuscript that addresses the points raised during the review process.

**Please consider both reviewer comments in full, and make the changes where needed or give a suitable response to why it cannot be completed - specifically in terms of quantitative methods.**

We look forward to receiving your revised manuscript.

Kind regards,

Emily Lowthian

Academic Editor

PLOS ONE

“Funding was received from the UKRI ESRC REF: ES/W001829/1.”

“Funding was received from the UKRI ESRC REF: ES/W001829/1. For the purpose of open access, the author has applied a Creative Commons Attribution (CC BY) licence to any Author Accepted Manuscript version arising from this submission.”

“Funding was received from the UKRI ESRC REF: ES/W001829/1.”

4. We note that this data set consists of interview transcripts. Can you please confirm that all participants gave consent for interview transcript to be published?

If they DID provide consent for these transcripts to be published, please also confirm that the transcripts do not contain any potentially identifying information (or let us know if the participants consented to having their personal details published and made publicly available). We consider the following details to be identifying information:

- Names, nicknames, and initials

- Age more specific than round numbers

- GPS coordinates, physical addresses, IP addresses, email addresses

- Information in small sample sizes (e.g. 40 students from X class in X year at X university)

- Specific dates (e.g. visit dates, interview dates)

- ID numbers

Or, if the participants DID NOT provide consent for these transcripts to be published:

- Provide a de-identified version of the data or excerpts of interview responses

- Provide information regarding how these transcripts can be accessed by researchers who meet the criteria for access to confidential data, including:

a) the grounds for restriction

b) the name of the ethics committee, Institutional Review Board, or third-party organization that is imposing sharing restrictions on the data

c) a non-author, institutional point of contact that is able to field data access queries, in the interest of maintaining long-term data accessibility.

d) Any relevant data set names, URLs, DOIs, etc. that an independent researcher would need in order to request your minimal data set.

For further information on sharing data that contains sensitive participant information, please see: https://journals.plos.org/plosone/s/data-availability#loc-human-research-participant-data-and-other-sensitive-data

If there are ethical, legal, or third-party restrictions upon your dataset, you must provide all of the following details (https://journals.plos.org/plosone/s/data-availability#loc-acceptable-data-access-restrictions):

1. A complete description of the dataset

2. The nature of the restrictions upon the data (ethical, legal, or owned by a third party) and the reasoning behind them

3. The full name of the body imposing the restrictions upon your dataset (ethics committee, institution, data access committee, etc)

4. If the data are owned by a third party, confirmation of whether the authors received any special privileges in accessing the data that other researchers would not have

5. Direct, non-author contact information (preferably email) for the body imposing the restrictions upon the data, to which data access requests can be sent

Reviewers' comments:

Reviewer's Responses to Questions

**Comments to the Author**

1. Is the manuscript technically sound, and do the data support the conclusions?

Reviewer #1: Yes

Reviewer #2: Partly

2. Has the statistical analysis been performed appropriately and rigorously? 

Reviewer #1: Yes

Reviewer #2: No

3. Have the authors made all data underlying the findings in their manuscript fully available?

Reviewer #1: Yes

Reviewer #2: Yes

4. Is the manuscript presented in an intelligible fashion and written in standard English?

Reviewer #1: Yes

Reviewer #2: Yes

5. Review Comments to the Author

Reviewer #1: This study is an excellent example of how qualitative data when properly integrated and mapped can provide comprehensive pathway in understanding unique lived experiences. Moreover, the study further demonstrates value of mapping individual experiences into a wider context and how such approaches can support development of tools that can be easily accessible for target population to 1) prepare 2)support and 3)guide the experience.

Reviewer #2: The paper describes the acceptability testing of the Carers ID intervention to support the mental health of family carers of people with profound and multiple intellectual disabilities. I found the paper easy to read and quite straightforward in the flow through the story of the paper. I appreciate the truly mixed methods approach to the assimilation of the qualitative and quantitative data - this is refreshing to see within studies and not often approached this robustly. On the whole I liked this manuscript and would support the publication of this.

There are a few areas that I think would benefit from some additional justification or clarification:-

It is noted that the acceptance scale has been adapted from another measure. Whilst some psychometrics are provided for this adaptation the description and process of this adaptation requires further justification given that it looks like these values have been calculated on the collected data that is to be analysed. Confirmation of data structure and measure validity would have been better conducted on an independent data set and possible introduction of any bias should be considered by conducting confirmatory psychometric analysis alongside the analysis of the data.

Participants are rewarded for their time to the tune of £10 and £20 - it is not clear when these are received or in what they relate to in terms of the timeline. For reproducibility it would be better to understand the flow of these rewards within the research scope and timeline.

The data analysis section is worded strangely - "descriptive statistics were used to calculate means and standard deviations for each subscale and a total score on the AHAA" This implies that subscales and scores were broken down by descriptive categories (which are not presented). The description of analysis could be clearer

"Participants could score a minimum of 22 or a maximum of 88 with higher scores indicating a greater degree of acceptability" This sentence was repeated 3 times in the manuscript, I would suggest avoiding repetitive use of information.

Sample size justification - there was no justification that the sample size for the quantitative elements were adequate. Given the design there may not have been a power calculation as such but there should be some justification of the sample aimed for or of that captured in the ability of the sample to answer the research aims and objectives.

Table 4 provides some of the descriptive information of the measures - It appears there may be some skew within the data (and possibly ceiling effects on at least the first three subscales), therefore I wonder would it be more appropriate to include median and IQR as well, ensuring that the data is being represented accurately.

I'm not convinced of the added value of the t-tests conducted - what were the hypothesis here, this was not outlined as part of the original research questions or part of the aims and objectives and with no justification on sample achieved it is difficult to tell if these would be expected findings or those generated by a small sample, particularly given the imbalance seen within some of the groups. Additionally, given the potential skew seen in the data did the required assumptions for the tests hold? Possibly providing the breakdown of the scores by comparison group in a supplementary table would enable the readers to understand the data presented more thoroughly.

Finally, there still appears to be some question over the effectiveness of this intervention. I acknowledge that this was not an aim or objective of this piece of research but given the findings I think it would be useful in the discussion to expand upon future plans for understanding the effectiveness of the intervention and what that may entail.

6. PLOS authors have the option to publish the peer review history of their article (what does this mean?). If published, this will include your full peer review and any attached files.

Reviewer #1: **Yes: **Milica Petrovic

Reviewer #2: **Yes: **Dr. Z. Hoare

---

## [Author Response · Author response to Decision Letter 0]

13 Aug 2024

The authors would like to thank the reviewers for their comments. We have addresses all points that the reviewers have raised.

---

## [Editor Report · Decision Letter 1]

15 Aug 2024

PONE-D-24-12377R1Acceptability testing of the Carers-ID intervention to support the mental health of family carers of people with profound and multiple intellectual disabilities.PLOS ONE

Dear Dr. Leonard,

Thank you for submitting your manuscript to PLOS ONE. After careful consideration, we feel that it has merit but does not fully meet PLOS ONE’s publication criteria as it currently stands. Therefore, we invite you to submit a revised version of the manuscript that addresses the points raised during the review process.

Please can the authors supply a point-by-point response to reviewers document or write-up on the platform. This is so reviewers can see how each comment has been considered - both positive and constructive. 

We look forward to receiving your revised manuscript.

Kind regards,

Emily Lowthian

Academic Editor

PLOS ONE

**Additional Editor Comments:**

Hi there,

Thank you for revising the paper. Please could you provide a full response to the reviewers, point-by-point explaining how you have addressed their comments so they can re-review.

Kind regards,

Emily Lowthian

---

## [Author Response · Author response to Decision Letter 1]

23 Aug 2024

Please see document - Response to Reviewers.

This is a point by point response to all reviewers comments

---

## [Decision Letter · Decision Letter 2]

11 Sep 2024

PONE-D-24-12377R2Acceptability testing of the Carers-ID intervention to support the mental health of family carers of people with profound and multiple intellectual disabilities.PLOS ONE

Dear Dr. Leonard,

Thank you for submitting your manuscript to PLOS ONE. After careful consideration, we feel that it has merit but does not fully meet PLOS ONE’s publication criteria as it currently stands. Therefore, we invite you to submit a revised version of the manuscript that addresses the points raised during the review process.

Thank you for your resubmission. Given reviewer one originally accepted, we have not sought further re-review. Reviewer 2 has accepted the revision with some optional comments you may want to consider.

As the Editor, I was checking your submission against the guidelines. I am keen to accept this submission providing some very minor amendments can be noted:

- The ethics statement should note on the manuscript that informed written consent was gathered.

- The limitations should consider the small sample size for the quantitative work, and any limitations with confounding in analysis.

If these are progressed, I am happy to move this manuscript to accept.

We look forward to receiving your revised manuscript.

Kind regards,

Emily Lowthian

Academic Editor

PLOS ONE

Journal Requirements:

Reviewers' comments:

Reviewer's Responses to Questions

**Comments to the Author**

1. If the authors have adequately addressed your comments raised in a previous round of review and you feel that this manuscript is now acceptable for publication, you may indicate that here to bypass the “Comments to the Author” section, enter your conflict of interest statement in the “Confidential to Editor” section, and submit your "Accept" recommendation.

Reviewer #2: All comments have been addressed

2. Is the manuscript technically sound, and do the data support the conclusions?

Reviewer #2: Yes

3. Has the statistical analysis been performed appropriately and rigorously? 

Reviewer #2: Yes

4. Have the authors made all data underlying the findings in their manuscript fully available?

Reviewer #2: Yes

5. Is the manuscript presented in an intelligible fashion and written in standard English?

Reviewer #2: Yes

6. Review Comments to the Author

Reviewer #2: Thank you for addressing all of my comments. It is really refreshing to see a truly mixed methods approach to utilising qualiative and quantitative data rather than presenting simply stand alone results.

Minor comment "Participants who completed the survey received a £10 voucher following competition." - think this should read completion

Apologies if I had not made my comment previous comment clear "Finally, there still appears to be some question over the effectiveness of thisintervention. I acknowledge that this was not an aim or objective of this piece of research but given the findings I think it would be useful in the discussion to expand upon future plans for understanding the effectiveness of the intervention and what that may entail." I understand that assessing effectiveness was not an aim of the study however, I wondered whether in the discussion it would be useful to include what future plans would be for establishing effectiveness of the intervention given you have established acceptability here but this would be merely a preference inclusion rather than a requirement.

7. PLOS authors have the option to publish the peer review history of their article (what does this mean?). If published, this will include your full peer review and any attached files.

Reviewer #2: No

---

## [Author Response · Author response to Decision Letter 2]

16 Oct 2024

Thank you for reviewing our revisions. We have addressed both comments from the reviewers fully - please see response to reviewers document for details.

---

## [Editor Report · Decision Letter 3]

18 Oct 2024

Acceptability testing of the Carers-ID intervention to support the mental health of family carers of people with profound and multiple intellectual disabilities.

PONE-D-24-12377R3

Dear Dr. Leonard,

We’re pleased to inform you that your manuscript has been judged scientifically suitable for publication and will be formally accepted for publication once it meets all outstanding technical requirements.

Kind regards,

Emily Lowthian

Academic Editor

PLOS ONE

---

## [Editor Report · Acceptance letter]

23 Oct 2024

PONE-D-24-12377R3 

PLOS ONE

Dear Dr. Leonard, 

I'm pleased to inform you that your manuscript has been deemed suitable for publication in PLOS ONE. Congratulations! Your manuscript is now being handed over to our production team.

Kind regards, 

on behalf of

Dr. Emily Lowthian 

Academic Editor

PLOS ONE